# Relationship between Pulmonary Regurgitation and Ventriculo–Arterial Interactions in Patients with Post-Early Repair of Tetralogy of Fallot: Insights from Wave-Intensity Analysis

**DOI:** 10.3390/jcm11206186

**Published:** 2022-10-20

**Authors:** Nikesh Arya, Silvia Schievano, Massimo Caputo, Andrew M. Taylor, Giovanni Biglino

**Affiliations:** 1Faculty of Mathematical and Physical Sciences, University College London, London WC1E 6BT, UK; 2Institute of Cardiovascular Science, University College London, London WC1E 6BT, UK; 3Centre for Cardiovascular Imaging, Great Ormond Street Hospital for Children NHS Foundation Trust, London WC1N 3HJ, UK; 4Bristol Medical School, Faculty of Health Sciences, University of Bristol, Bristol BS8 1TH, UK; 5Bristol Heart Institute, University Hospitals Bristol & Weston NHS Foundation Trust, Bristol, BS2 8HW, UK; 6National Heart and Lung Institute, Imperial College London, London SW7 2BX, UK

**Keywords:** tetralogy of fallot, congenital heart disease, left ventricle, aorta, wave-intensity analysis, magnetic resonance imaging, cyanosis, pulmonary regurgitation

## Abstract

This study aimed to investigate the effect of pulmonary regurgitation (PR) on left ventricular ventriculo–arterial (VA) coupling in patients with repaired tetralogy of Fallot (ToF). It was hypothesised that increasing PR severity results in a smaller forward compression wave (FCW) peak in the aortic wave intensity, because of right-to-left ventricular interactions. The use of cardiovascular magnetic resonance (CMR)-derived wave-intensity analysis provided a non-invasive comparison between patients with varying PR degrees. A total of n = 201 patients were studied and both hemodynamic and wave-intensity data were compared. Wave-intensity peaks and areas of the forward compression and forward expansion waves were calculated as surrogates of ventricular function. Any extent of PR resulted in a significant reduction in the FCW peak. A correlation was found between aortic distensibility and the FCW peak, suggesting unfavourable (VA) coupling in patients that also present stiffer ascending aortas. Data suggest that VA coupling is affected by increased impedance.

## 1. Introduction

Arterial pulse wave transmission through the vasculature depends on numerous factors, such as myocardial function and vessel wall stiffness. Analysis of wave propagation is, thus, of clinical interest, as it holds information regarding cardiac function. There are a number of ways to evaluate the wave propagation; one such is wave-intensity analysis (WIA). The standard definition of wave intensity is the product of simultaneous pressure and velocity differentials at one location in the vasculature [1].

Wave-intensity analysis is a hemodynamic index yielding information on the working condition of the heart in relation to the state of the vasculature. In clinical terms, this relationship is described by ventriculo–arterial (VA) coupling measurements, typically evaluated using effective arterial elastance (Ea, which encompasses the vascular resistance, compliance and systolic and diastolic time intervals as an index of the afterload) and maximal systolic ventricular elastance (Emax), which is a load-independent index (i.e., ratio of mean arterial pressure to end-systolic volume). It is commonly accepted that optimal VA coupling occurs when Ea/Emax = 1.

The clinical applicability of WIA, historically, has been limited by the invasive nature of the pressure data required to perform the analysis. Feng and Khir (2010) used theoretical analyses to calculate wave speed and wave separation non-invasively using vessel diameter and blood velocity [2]. More recently, formulating wave intensity in terms of the product of simultaneous changes in velocity and fractional changes in the area rendered the analysis non-invasive and suitable for MRI data, without the assumption of vessel circularity [3].

Wave-intensity analysis has been shown to produce valuable insights into scenarios of repaired congenital heart disease (CHD) [4], where VA coupling can be negatively affected, both on the “V” and the “A” side. Here, we sought to apply the analysis to the case of tetralogy of Fallot (ToF). ToF is the most common cyanotic CHD, the cyanosis being proportional to degree of right ventricular outflow tract (RVOT) obstruction. Historically, surgical palliation relied on blood redirection methods (e.g., Blalock–Taussig, BT, or Waterston shunts). Lower risk procedures are now used to treat ToF, mainly through ventricular septal defect (VSD) patch closure and RVOT relief. Such procedures also reduce the risk of long-term complications. However, pulmonary regurgitation (PR) is still an avid issue that, depending on the severity, can comprise cardiac function in the patient’s future and lead to right ventricular dysfunction, a large cause of late morbidity and mortality in these patients [5]. The determinants of PR can be anatomic or hemodynamic factors such as a pulmonary valve abnormality (hypolastic or absent bicuspid), repair of pulmonary stenosis with a transannular patch or pulmonary perfusion abnormalities [6]. The pathophysiological significance of PR stems from the adaptation of the right ventricle (RV) in response to the PR and, therefore, this depends on the duration and degree of PR. PR results in RV volume overload, with an increased end-diastolic volume (EDV).

In patients with repaired ToF, PR occurs as a result of compromised cardiovascular mechanisms in the right side of the heart. The increase in RV compliance leads to an increase in end-systolic volume. Following surgical palliation of ToF, PR can occur at different levels of severity. Applying WIA will allow us to assess the VA coupling of a group of patients with differing degrees of PR, aiming to elucidate whether VA coupling efficiency is affected by differing left–right interactions impacting cardiovascular function, stemming from increasing amounts of PR. Indeed, LV and RV function are interrelated in ToF [7], with RV mechanics inducing geometric changes in the LV linked to both diastolic and systolic mechanical inefficiency [8,9] and RV dilation perturbing presystolic LV flow patterns and negatively affecting LV performance, by way of a leftward septal shift in diastole [8,9]. Reduced contractile function in relation to the degree of PR has been suggested [10]. We, therefore, aimed to see whether the application of WIA in this context could reveal mechanical insight regarding the changes in LV VA coupling.

We hypothesised that more severe PR results in reduced left ventricular contractility, due to the larger extent of right–left interactions, in turn impinging on the coupling with the aorta. We specifically hypothesised lower forward compression wave intensity/energy with higher PR severity.

## 2. Materials and Methods

### 2.1. Patient Population

This was a retrospective observational study on patients with repaired tetrology of Fallot. A list of n = 1077 patients was collated from the Great Ormond Street Hospital Clinical Database. The main exclusion criteria were: patients who had the disease repaired with the Blalock–Taussig shunt method; patients repaired after one year of age; poor image quality; patients who already had a percutaneous pulmonary valve implantation; and incomplete data (accounting for n = 223). The choice of <1 year age of repair was based on creating a more homogenous study population, particularly with reference to the literature indicating that ToF patients with primary repair at 3–11 months have better survival and physiological outcomes [11,12], including a recent systematic review analysing at a total of 143 papers of 137 distinct cohorts comprising 21,427 patients [13]; in addition, <1 year age of repair is representative of our Centre’s population, where in asymptomatic patients, elective primary repair is undertaken generally around 6 months [14].

The resulting study population was n = 201 patients, categorized into: 0% PR (n = 17), mild PR (1–19%, n = 38), moderate PR (20–39%, n = 90), severe PR (>40%, n = 56). Conventional demographics and hemodynamic parameters were collected from the CMR report, and indexed when appropriate.

### 2.2. CMR Imaging

All patients had undergone CMR imaging as part of their clinical examination. Aortic flow quantification was performed using phase-contrast sequences. Processing of the images was carried out using plug-ins written in-house (OsiriX, Pixmeo, Bernex, Switzerland). Segmentation of the aorta was performed on the modulus image, using a validated semi-automatic registration-based algorithm [15]. The ability to alter the regions of interest (ROI) was available to allow for the refinement of vessel wall delineation. The final sets of ROIs were used for the calculation of the aortic cross-sectional area (A) and the mean aortic velocity (U) from the phase images, which in turn allowed the calculation of wave speed, *c*, and wave energy, *I*.

Of the 201 patients, 110 (n = 9 with no PR; n = 16 with mild PR; n = 52 with moderate PR; and n = 33 with severe PR) had right branch bundle block due to a conduction issue altering the electrical activity of the heart, reflecting on the velocity signals measured with CMR. These velocity profiles were computationally adjusted during post-processing in order to rectify the velocity curve for the purpose of running the analysis. Adjusted profiles were not included for wave-speed observations but were included for wave-intensity observations.

### 2.3. Wave-Intensity Analysis

For the full derivation of the wave-intensity formulation, we refer the reader to Parker’s review covering key nomenclature and mathematical derivations [16]. The area-based wave-intensity (dI_A_) [17] pattern allows us to identify the type of wave and classify them as either compression (dlnA > 0) or expansion (dlnA < 0) waves. Waves can also be classified as forward- or backward-travelling (i.e., away or toward the heart). Traditionally, a forward compression wave (FCW) is associated with LV ejection and a forward expansion wave (FEW) with LV relaxation. Ohte et al. (2003) showed that in healthy subjects, the FCW significantly correlates with the maximum rate of pressure rise (dP/dt), and the FEW correlates with the LV diastolic time constant. The peak intensity of the FCW and FEW and the corresponding energy carried by such waves were parameters of interest in this analysis. Wave energy (I) was derived from the area under the wave-intensity curve.

The ascending aortic CMR flow data were segmented using a validated registration-based segmentation algorithm which automatically extracts U and A signals. According to the water hammer equation, the lnA-U relationship allows for the derivation of the wave speed (c) by taking the linear slope in early systole (i.e., when no reflected waves are expected). The aortic lnA-U relationship is depicted by a loop. The wave speed, in turn, allows the derivation of aortic distensibility D by using the Bramwell–Hill equation:D = 1/(ρc^2)
where ρ is the blood density.

### 2.4. Data Analysis and Statistics

A typical dI_A_ pattern is observed in the patients with repaired ToF. The peaks and areas of the FCW and FEW waves were measured from the net dI_A_ plots.

Data are presented as mean ± standard deviation, counts or proportions. A chi-squared test was carried out to test differences in categorical variables. Differences across the four groups as defined by PR severity were assessed with the Kruskal–Wallis test with Dunn’s test post hoc. Linear regression analysis and multivariable regression analysis were performed to establish the relationship between some of the clinical and wave-intensity parameters. A value of *p* < 0.05 was considered statistically significant.

With regard to statistical power, data on ToF are currently not available, so we considered another scenario of cyanotic CHD, i.e., transposition of the great arteries, for which wave-intensity pilot data were available [3]. For peak FCW, with a sample of 7 subjects per group, we had 90% of power to detect a difference of 4.2 × 10–5 m/s between the null hypothesis (i.e., no difference between TGA and controls) and the alternative hypothesis (i.e., expecting a difference between in mean FCW magnitude of 6.3 × 10^−5^ and 2.1 × 10^−5^ m/s with standard deviations of 2.2 × 10^−5^ and 1.5 × 10^−5^ m/s, respectively), at a significance level of 0.05. Considering peak FEW, with a sample of 18 subjects per group, we had 90% of power to detect a difference of 0.5 × 10^−5^ between the null hypothesis (i.e., no difference between TGA and controls) and the alternative hypothesis (i.e., expecting a difference in mean FEW magnitude of 0.8 × 10^−5^ and 0.3 × 10^−5^ m/s with standard deviations of 0.5 × 10^−5^ and 0.2 × 10^−5^ m/s, respectively), at a significance level of 0.05.

## 3. Results

### 3.1. Ventricular Function from CMR

Subjects with no PR had a significantly lower EDV (88 ± 17 mL/m^2^) than subjects with severe PR (154 ± 35 mL/m2, *p* < 0.001) and ESV (no PR: 38 ± 14 mL/m^2^, vs. severe PR: 73 ± 27 mL/m^2^, *p* < 0.001). The right ventricular stroke volume thus shows significant changes (0% PR: 83 ± 15 mL/beat, vs. severe PR: 114 ± 42 mL/beat, *p* < 0.001). Further marked differences in indexed cardiac output were observed. LV ejection fraction showed a significant reduction from no PR to severe PR (67 ± 7% vs. 61 ± 6%, *p* = 0.002). Results are summarized in Table 1.

### 3.2. Wave-Intensity Analysis

Overall, the FCW peak and area were reduced significantly with increasing PR (*p* = 0.015 and *p* = 0.021, respectively), differences predominantly being driven by the no PR group (Figure 1). There were no significant differences between the groups for the FEW peak and area. Results are summarized in Table 2.

Regression analysis showed significant but weak correlations between the FCW peak (Figure 2) and extent of PR (R^2^ = 0.053, *p* = 0.001), age (R^2^ = 0.037, *p* = 0.006) and aortic distensibility (R^2^ = 0.073, *p* = 0.001). On the multivariate regression including age, sex, PR% and distensibility (R^2^ = 0.1), it was observed that only distensibility was a statistically significant predictor of the FCW peak (*p* = 0.003), as reported in Table 3.

Significant differences were measured in terms of wave speed (Figure 3), which increased significantly with increasing PR (from 3.9 ± 0.7 m/s for the 0% PR group to 5.8 ± 1.5 m/s, *p* < 0.001, for the severe PR group). This reflects a significant reduction in aortic distensibility between the groups.

## 4. Discussion

In this study, we have shown how non-invasive area and velocity measurements can be applied to the clinical scenario of tetralogy of Fallot, where we have analysed the implications of pulmonary regurgitation on patients with repaired ToF.

This retrospective study focused on the interactions between systemic ventricle and aorta within four groups according to the amount of PR, based on the hypothesis that increasing PR would reflect as a reduction in the peak FCW, indicating unfavourable VA coupling. This hypothesis was confirmed whereby a significant reduction in the peak FCW was measured for patients with no PR compared to those with any PR. Differences between the mild, moderate and severe PR groups instead were not statistically significant, suggesting the absence of a progressive effect. Furthermore, regression analysis suggested that changes in FCW are significantly associated with aortic distensibility, rather than PR, and indeed, we observed decreasing distensibility with increasing PR severity in this cohort. We can infer from these initial results that any patient with repaired ToF with PR will have somewhat unfavourable VA coupling compared to a patient with repaired ToF and with a competent pulmonary valve.

Past studies have shown that PR and its severity influences physiological implications including exercise tolerance, incidence of atrial and ventricular arrhythmia, and the risk of unexpected cardiac death [18,19]. Interventional pulmonary valve replacement (PVR) has shown to improve ventricular function and reduce atrial and ventricular arrhythmias [20,21,22]. Similar observations have been carried out on the right side of the heart whereby the pulmonary artery has been assessed. In our study, we were primarily interested in analysing aortic data in order to assess systemic VA coupling and the effect of right–left interactions on this parameter. From a methodological standpoint, if we were to use the same WIA method on the right side, then the analysis of CMR flows in the pulmonary artery would likely suffer from an out-of-plane motion, and this should be taken into account when planning the analysis [23].

The benefit of carrying out CMR-derived WIA is the resulting insight it provides into VA coupling mechanisms. The FCW and FEW have been linked to ventricular performance [4]. Focusing on the FCW, a decrease in its amplitude suggests reduced ventricular contractility. Indeed, this can provide additional physiological insight on potential subclinical changes, including an effect on systolic function in the face of normal EF. Our results, indeed, showed reduced FCW with any degree of PR, and furthermore, the regression analysis suggested a significant association with aortic distensibility. We can, thus, infer that aortic distensibility likely plays a role in the modulation of the FCW peak and ventricular contractility, as reflected by our wave-speed results (i.e., increasingly reduced distensibility with increasing PR severity).

Significant differences in wave speed were measured between the groups. This increase in wave speed is reflected in the patients having a lower aortic distensibility. Grotenhuis et al. (2009) have previously shown similar results of wave speeds experienced in ToF patients reporting velocities of similar ranges, namely 5.5 m/s vs. 4.6 m/s (ToF patients with pulmonary valve replacement vs. healthy controls). This reduction suggests that increasing PR% is accompanied by stiffening of the aorta, in agreement with previous observations [24]. The effect of distensibility is interesting as it may be a more profound reason of LV burdening than that possibly caused by right-to-left interactions, in agreement with the wave-intensity results.

Right ventricular dilatation and reduced RV function in patients after the repair of ToF have been established in past studies [25]. Our hemodynamic CMR data highlighted a number of significant differences in the right ventricle. The EDVi, ESVi, SV and CI all increase with the increased severity of PR. These findings agree with findings in the literature, both in invasive [26] and CMR studies [27].

In the systemic ventricle, differences between patients with 0% and severe PR were observed in terms of LVEF and SV, both reduced in patients with severe PR. Previous studies have observed reduced LVEF in ToF patients, however, the majority of ToF patients have preserved LVEF [28]. Correlations between left and right ventricular dysfunction on CMR have also been previously reported [28].

Considering that the analysis has suggested a key role of aortic distensibility and that differences in aortic stiffness have been identified, future work should complement observations on distensibility also with aortic dimensions and perhaps overall morphology. This could involve a statistical shape modelling approach to assess the aortic arch in its three-dimensionality, as already explored in other CHD scenarios such as bicuspid aortic valve aortopathy [29,30]. Aortic root pathology has been reported in ToF, with one multicentric echocardiographic study [31] noting that one third of patients with repaired ToF had aortic root diameters of >40 mm, yet the prevalence of root dilation when defined by an indexed ratio of observed/expected values was low. This is an interesting point for future investigation.

### 4.1. Limitations

This study suffers from the limitations of a retrospective cross-sectional study design. Wave speed and distensibility are widely noted as age-dependent parameters [32], and whilst a small mean age difference particularly between the 0% and sever PR groups may represent a confounding factor, this is arguably attenuating differences in this case (it is reasonable to assume that if severe PR patients were older, their wave speed would be even higher). Those patients who presented branch-bundle-block conductance were excluded from wave-speed analysis, although we noted that they unlikely skewed the results, as they were spread across all four groups. The out-of-plane motion may pose an issue in the segmentation of scans and give rise to inaccurate area estimates, particularly if this analysis was repeated on the right side. Previous studies have quantified this movement on the aortic side to be less than 1 cm [33,34], and so area measurements should remain largely unaffected, especially as regional wall properties remain constant. Finally, from a methodological standpoint, this study used retrospective data, and the CMR data are of Cartesian type (~30 ms temporal resolution). Whilst net-wave-intensity calculations are not affected, temporal resolution can affect the calculation of the wave speed [35], and a higher temporal resolution would be desirable for future prospective studies. Despite this limitation, the measured wave-speed values are in agreement with figures from previous CMR literature using the foot-to-foot method [36].

### 4.2. Clinical Significance

The benefit of non-invasive CMR-derived wave-intensity analysis lies in the potential additional nuanced insight into the working condition of the heart. The simplicity of the segmentation and derivation of the wave-intensity data has the potential of clinical applicability and integration in clinical analyses, with the advantage of being more intuitive than classical wave separation [37].

Previous studies have focused on patients with severe PR in understanding their pathophysiological implications or evaluating their need for pulmonary valve replacement (PVR) [38] and other interventions. For example, Frigiola et al. (2008) assessed the timing of PVR, focusing on patients with severe PR. Based on our results, it would be interesting to further investigate differing severities of PR for left heart function prospectively. In particular, it may be interesting to assess other signs that may suggest a dysfunction of the LV despite the preserved EF. While this is only a preliminary study, it reinforces the need for a nuanced appreciation of both LV and RV mechanics in relation to tailoring PVR in TOF patients.

## 5. Conclusions

Wave-intensity analysis derived from CMR data was successfully applied to a cohort of patients with tetralogy of Fallot, observing that the reduced forward compression wave amplitude was associated with increasing aortic distensibility, suggesting unfavourable VA coupling in patients with stiffer aortas. Future work may develop wave intensity as an index offering (i) additional insights into subclinical changes in CHD patients and (ii) nuanced information regarding ventricular energetics, contributing to refined clinical decision-making.

## Figures and Tables

**Figure 1 jcm-11-06186-f001:**
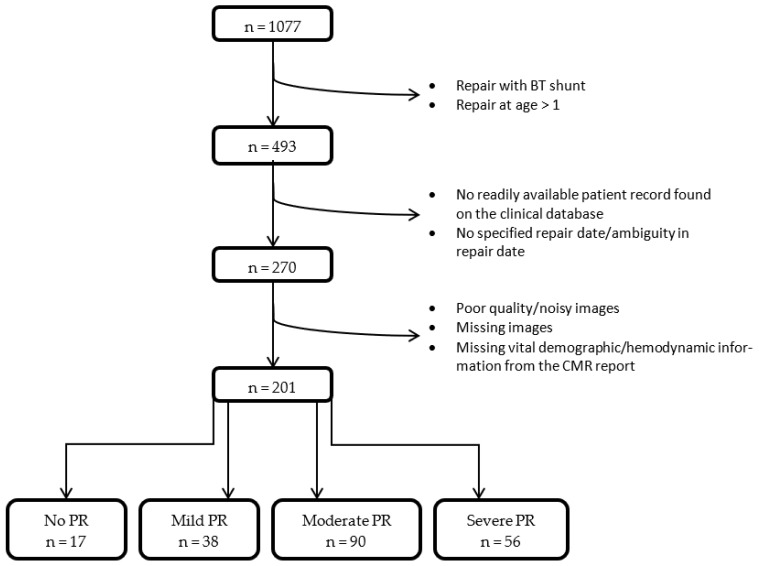
Flow chart of patient selection. BT = Blalock–Taussig, CMR = cardiovascular magnetic resonance, PR = pulmonary regurgitation.

**Figure 2 jcm-11-06186-f002:**
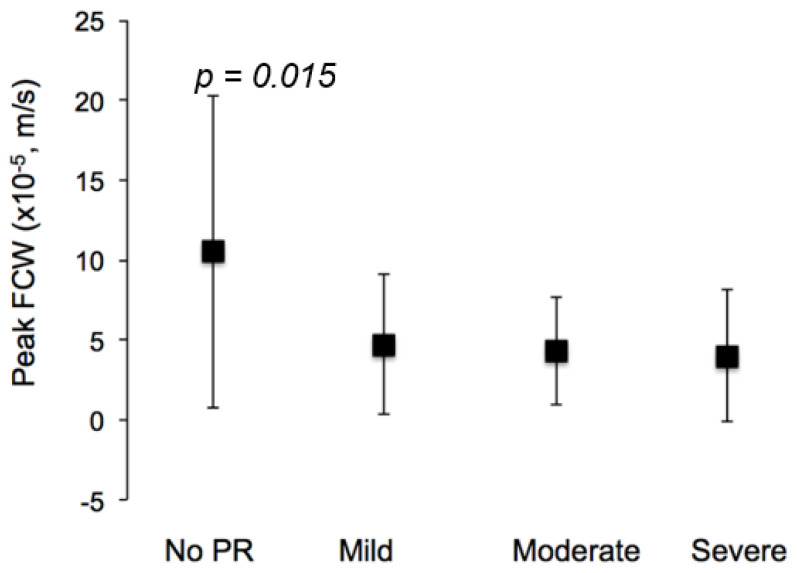
ToF−repaired patients exhibited significant differences in FCW peaks between patients with no PR and those with any severity of PR. This change was also seen in the wave energy (taken from the FCW area).

**Figure 3 jcm-11-06186-f003:**
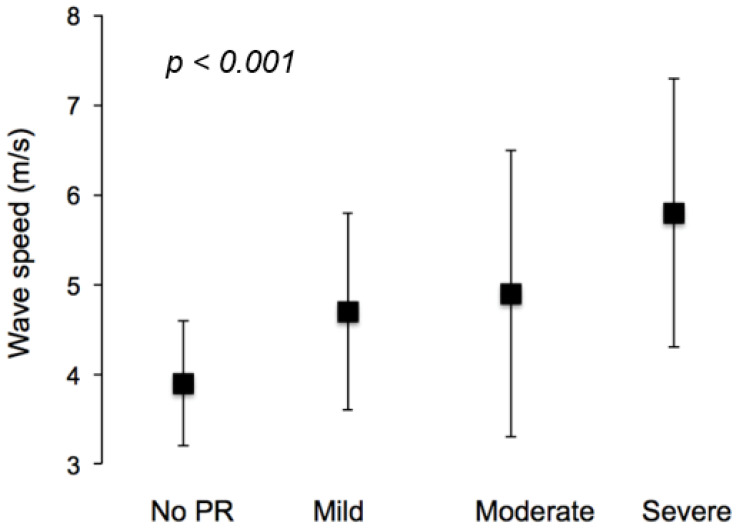
ToF-repaired patients exhibited increasing wave speed with increasing PR severity.

**Table 1 jcm-11-06186-t001:** Summary of demographic data and CMR data comparing groups of patients with four differing levels of severity of pulmonary regurgitation.

Variable	No PR (n = 17)	Mild (n = 38)	Moderate (n = 90)	Severe (n = 56)	*p* (Overall)
Age at magnetic resonance imaging (MRI) follow-up scan (y)	17 ± 5	16 ± 10	15 ± 9	14 ± 6	0.062
Sex (M/F)	7/10	24/14	55/35	43/13	0.043
BSA (m^2^)	1.7 ± 0.3	1.5 ± 0.5	1.5 ± 0.4	1.4 ± 0.4	0.007
HR (bmp)	75 ± 10	78 ± 5	80 ± 13	79 ± 3	0.583
PR (%)	0	9 ± 6	30 ± 5	49 ± 6	0.000
LV EDVi (mL/m^2^)	73 ± 12	75 ± 11	72 ± 12	73 ± 17	0.331
LV ESVi (mL/m^2^)	25 ± 8	28 ± 10	26 ± 7	30 ± 9	0.064
LV SV (mL)	82 ± 13	74 ± 28	67 ± 21	64 ± 26	0.007
LV EF (%)	67 ± 7	64 ± 8	64 ± 5	61 ± 6	0.001
LV CI (L/min/m^2^)	3.6 ± 0.6	3.7 ± 0.7	3.6 ± 0.7	3.6 ± 1.0	0.733
RV EDVi (mL/m^2^)	88 ± 17	98 ± 26	112 ± 35	154 ± 35	0.000
RV ESVi (mL/m^2^)	38 ± 14	43 ± 15	49 ± 24	73 ± 27	0.000
RV SV (mL)	83 ± 15	83 ± 39	94 ± 33	114 ± 42	0.001
RV EF (%)	58 ± 10	59 ± 10	58 ± 9	54 ± 8	0.103
RV CI (L/min/m^2^)	3.7 ± 0.7	4.4 ± 1.0	5.1 ± 1.0	6.4 ± 1.4	0.000

**Table 2 jcm-11-06186-t002:** Summary of wave-intensity and wave-speed results. FCW = forward compression wave, FEW = forward expansion wave, PR = pulmonary regurgitation.

Variable	No PR (n = 17)	Mild (n = 38)	Moderate (n = 90)	Severe (n = 56)	*p* (Overall)
Wave speed (m/s)	3.9 ± 0.7	4.7 ± 1.1	4.9 ± 1.6	5.8 ± 1.5	0.000
Distensibility (×10^−3^ 1/mmHg)	8.8 ± 2.8	6.5 ± 3.0	6.8 ± 3.6	4.8 ± 3.0	0.000
FCW peak (×10^−5^, m/s)	10.5 ± 9.7	4.7 ± 4.4	4.3 ± 3.4	4.0 ± 4.1	0.015
FCW area (×10^−3^, m)	2.7 ± 2.1	1.4 ± 1.3	1.3 ± 1.0	1.3 ± 1.3	0.021
FEW peak (×10^−5^, m/s)	0.6 ± 0.4	0.6 ± 0.5	0.7 ± 0.6	0.6 ± 0.4	0.578
FEW area (×10^−3^, m)	0.3 ± 0.2	0.3 ± 0.2	0.3 ± 0.2	0.3 ± 0.2	0.415

**Table 3 jcm-11-06186-t003:** Summary of the regression analysis for predictors of the forward compression wave (FCW). PR = pulmonary regurgitation, MRI = magnetic resonance imaging.

FCW Predictor	Univariate	Multivariate	
Extent of PR (%)	*p* = 0.001, R^2^ = 0.053	*p* = 0.971	
Age at time of MRI (y)	*p* = 0.006, R^2^ = 0.037	*p* = 0.065	R^2^ = 0.097
Sex (M/F)	*p* = 0.878, R^2^ = 0.000	*p* = 0.948	
Distensibility	*p* = 0.001, R^2^ = 0.073	*p* = 0.003	

## Data Availability

Institutional ethical approval was obtained for the retrospective use of the CMR data. The parents of all patients gave informed consent for research on the use of these imaging data.

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
