# Peer review of "Relationship between Pulmonary Regurgitation and Ventriculo–Arterial Interactions in Patients with Post-Early Repair of Tetralogy of Fallot: Insights from Wave-Intensity Analysis"

_jcm, 2022, doi:10.3390/jcm11206186_

Round 1

Reviewer 1 Report

The authors studied wave intensity analysis in repaired tetralogy of Fallot is a novel idea to assess the VA coupling.

However, there are several comments worth to consider:

1/ More description of principle of wave intensity analysis with respect to haemodynamics is recommended to enable the reader to understand the concept and the interpretation of FCW and wave sepeed.

2/ There is lack of theoretical discussion on the hypothesis or rationale of their study that pulmonary regurgitation and LV VA coupling is related

3/ Exclusion criteria: this is not certain why the authors excluded the patients with repaired at age >1yo. The rationale of how this could influence the study was not described.

4/ The correlation between pulmonary regurgitation and FCW was not substantiated in multivariate analysis. And there was no correlation of FCW and the severity of pulmonary regurgitation. Thus, this proved that the relation of FCW and PR may not be materialistic enough.

5/ There was no information regarding the aortic root dimension. There have been extensive data showing aortopathy including aortic root dilation, altered aortic distensibility, and aortic stiffness in repaired TOF patients. And aortopathy may be more involved in altered LV VA coupling rather than pulmonary regurgitation, especially if taken the authors' data into acount. 

6/ Clinical implication of understanding FCW in the setting of repaired TOF with PR is not discussed enough. Is this implicated in LV dysfunction of repaired TOF, or as parameters to help determine the timing of intervention (e.g. PVR) in repaired TOF?

Author Response

We include a point-by-point response and revised manuscript 

Reviewer 2 Report

The article is well written. There are clear structure and description of the background, methods, study design and objectives.
The results are clearly described with an understandable table and figures.
The discussion, including study limitations, analyses results exhaustively and compares with other articles in this field.
The conclusions are clear and underline the innovations of the study and possible future work.

Author Response

(The authors gave the same response as above.)

Reviewer 3 Report

The investigation performed by Arya, et al showed a relationship between PR and VA coupling in children with TOF. They showed a how non-invasive area and velocity measurements can be applied to the clinical scenario of TOF. This investigation is interesting yet weakened by the flowing concerns.

1. Table 1, The first row, Age(y) should be Age(m), not “year” but “month”? Generally, in our hospital, we corrected TOF at age 3-6 month or around 1 year old; at the age of 14-year-old or above , it’s rare. It’s unclear how long the PR lasted. Does the duration of PR affect left and right heart function and FCW?

2. It’s unclear why the authors chose TOF patients whose defect had been repaired within one year post birth? Please explain.

3. In the introduction, the last paragraph, the authors hypothesise that more PR results in reduced contractility, of which? Right ventricle or left ventricle? Please make it clear.

4. Data concerning FCW area, and FEW peak and area are missing.

5. Please provide data of Regression analysis.

6. In the abstract, there were too much background introduction and too little data of results, making the abstract less informative. In my opinion, the first and second half sentence is not necessary.

7. How the authors make the conclusion that PR severity did not appear to burden the LV and its coupling with aorta? Table1 showed LV EF and SV, FCW peak were correlated with PR severity. In the discussion, the authors also mentioned that patients with PR have somewhat  unfavourable VA coupling. Please explain.

Author Response

(The authors gave the same response as above.)

Round 2

Reviewer 3 Report

The authors have addressed all my concerns. I have no further comments.